# SARS-CoV-2 Spike Mutations, L452R, T478K, E484Q and P681R, in the Second Wave of COVID-19 in Maharashtra, India

**DOI:** 10.3390/microorganisms9071542

**Published:** 2021-07-20

**Authors:** Sarah Cherian, Varsha Potdar, Santosh Jadhav, Pragya Yadav, Nivedita Gupta, Mousumi Das, Partha Rakshit, Sujeet Singh, Priya Abraham, Samiran Panda, NIC Team

**Affiliations:** 1ICMR-National Institute of Virology, Pune 411001, India; sarahcherian100@gmail.com (S.C.); potdarvarsha9@gmail.com (V.P.); smjniv@gmail.com (S.J.); hellopragya22@gmail.com (P.Y.); mosu39@gmail.com (M.D.); 2Indian Council of Medical Research, New Delhi 110029, India; drguptanivedita@gmail.com (N.G.); samiranpanda.hq@icmr.gov.in (S.P.); 3National Centre for Disease Control, New Delhi 110054, India; partho_rakshit@yahoo.com (P.R.); sujeet647@gmail.com (S.S.)

**Keywords:** SARS-CoV-2, India, Maharashtra, evolution, second wave, whole genomes, B.1.617.1, B.1.617.2, modeling

## Abstract

As the global severe acute respiratory syndrome coronavirus 2 (SARS-CoV-2) pandemic expands, genomic epidemiology and whole genome sequencing are being used to investigate its transmission and evolution. Against the backdrop of the global emergence of “variants of concern” (VOCs) during December 2020 and an upsurge in a state in the western part of India since January 2021, whole genome sequencing and analysis of spike protein mutations using sequence and structural approaches were undertaken to identify possible new variants and gauge the fitness of the current circulating strains. Phylogenetic analysis revealed that newly identified lineages B.1.617.1 and B.1.617.2 were predominantly circulating. The signature mutations possessed by these strains were L452R, T478K, E484Q, D614G and P681R in the spike protein, including within the receptor-binding domain (RBD). Of these, the mutations at residue positions 452, 484 and 681 have been reported in other globally circulating lineages. The structural analysis of RBD mutations L452R, T478K and E484Q revealed that these may possibly result in increased ACE2 binding while P681R in the furin cleavage site could increase the rate of S1-S2 cleavage, resulting in better transmissibility. The two RBD mutations, L452R and E484Q, indicated decreased binding to select monoclonal antibodies (mAbs) and may affect their neutralization potential. Further in vitro/in vivo studies would help confirm the phenotypic changes of the mutant strains. Overall, the study revealed that the newly emerged variants were responsible for the second wave of COVID-19 in Maharashtra. Lineage B.1.617.2 has been designated as a VOC delta and B.1.617.1 as a variant of interest kappa, and they are being widely reported in the rest of the country as well as globally. Continuous monitoring of these and emerging variants in India is essential.

## 1. Introduction

The severe acute respiratory syndrome coronavirus 2 (SARS-CoV-2) surface spike (S) protein mediates entry into host cells by binding to the host receptor angiotensin-converting enzyme 2 (ACE2) via its receptor-binding domain (RBD). Crystal structures of SARS-CoV-2 S protein or its RBD complexed with ACE2 from different hosts reveal that the RBD contains a core and a receptor-binding motif (RBM) which forms contacts with ACE2 [1]. A number of naturally selected mutations in the RBM have been shown to affect infectivity, human-to-human transmission, pathogenesis and immune escape [2].

As the global SARS-CoV-2 pandemic expands, genomic epidemiology and whole genome sequencing are being used to investigate the transmission and evolution. The expansion of the genome-wide diversity resulted in delineation of the viral strains into clades, lineages and sub-lineages. Clades G/GH/GR/GV/GRY as per the Global Initiative on Sharing All Influenza Data (GISAID) database (https://www.gisaid.org/, Accessed on 14 June 2021) [3] possess a common mutation D614G in the spike protein. The mutation was shown to result in significantly higher human host infectivity and better transmission efficiency for the virus [4,5]. Three recently emerged “variants of concern” (VOCs) of SARS-CoV-2 are GRY (formerly GR/501Y.V1)/B.1.1.7 (alpha), GH/501Y.V2/B.1.351 (beta) and GR/501Y.V3/P.1 (gamma) [6]. These variants are known to possess multiple mutations across the genome, including several in the S protein and its RBD, such as N501Y, E484K and K417N/T [5,6,7,8]. Multiple SARS-CoV-2 variants are now seen to be circulating globally.

The potential introduction and consequence of these emerging variants in the country is vital to support the public health response. The National Influenza Centre at the ICMR-National Institute of Virology, Pune, as an apex laboratory of the Indian Council of Medical Research (ICMR), has been continuously involved in SARS-CoV-2 diagnostics and monitoring the genomic evolution of this virus. In addition, the ICMR-NIV, Pune is also one among the ten national laboratories of the Indian SARS-CoV-2 Consortium on Genomics (INSACOG), Ministry of Health and Family Welfare, Government of India, catering to western India. Against the backdrop of the global emergence of VOCs and an upsurge in Maharashtra, a state in the western part of India, enhanced sequencing was undertaken. This was to identify possible new variants and analyze spike protein mutants, in particular to gauge the fitness of current circulating strains, using bioinformatics sequence and molecular structural approaches.

## 2. Materials and Methods

The state of Maharashtra (latitude 19°39′ N, longitude 75°18′ E) is located in west central India, in the northwestern part of the Indian subcontinent. Since the end of January 2021, a concentrated spurt in COVID-19 cases was noted in several districts of Maharashtra (Appendix A). As per the state government directives, samples from international travelers and 5% of surveillance samples from the positive cases with Ct value <25, including those from clusters, long haulers and mild/moderate/severe and deceased cases, were referred to the ICMR-NIV, Pune for whole genome sequencing. Nasopharyngeal swabs (n = 2987, 25 November 2020–30 May 2021) were processed for whole genome sequencing. 

Briefly, nucleic acid extraction was performed using 280 μL of each sample in duplicate by a Qiagen viral RNA extraction protocol and quantified RNA was further processed for template preparation using the Ion Chef System. Purified template beads were submitted to meta transcriptome next-generation sequencing (NGS) in the Ion S5 platform (ThermoFisher Scientific, Waltham, MA, USA) using an Ion 540™ chip and the Ion Total RNA-Seq kit v2.0, as per the manufacturer’s protocol (ThermoFisher Scientific, Waltham, MA, USA) [9]. Sequence data were processed using Torrent Suite Software (TSS) v5.10.1 (ThermoFisher Scientific, Waltham, MA, USA). Coverage analysis plugins were utilized to generate coverage analysis report for each of the samples. Reference-based reads gathering and assembly were performed for all the samples using Iterative Refinement Meta-Assembler (IRMA) [10]. A subset of the samples was also sequenced on an Illumina machine and analyzed using CLC Genomics Workbench version 20 (CLC, Qiagen, Hilden, Germany) as described elsewhere [11]. Of 2204 whole genomes obtained, 1791 were considered for different analyses based on coverage of ≥93% of the genome. Both eastern (n = 829) and western (n = 962) districts of the state were well represented. For each whole genome sequence, the GISAID clade assignment was done using CoVsurver: Mutation Analysis of hCoV-19 (https://www.gisaid.org/epiflu-applications/covsurver-mutations-app/, Accessed on 1 June 2021). For lineage assignment, a web application, i.e., Phylogenetic Assignment of Named Global Outbreak LINeages (PangoLIN) COVID-19 Lineage Assigner (https://pangolin.cog-uk.io/, Accessed on 1 June 2021), was implemented [8]. The 1791 genomes were aligned using MAFFT v.7.450 [12] and a neighbor-joining phylogenetic tree was constructed using MEGA V.6 [13], employing the composite likelihood as the substitution model and 1000 bootstrap replications. The gene-wise top mutations in the predominant lineages were depicted using heat maps based on the calculations of the mutational frequencies.

For further structural characterization of the S protein mutations, the crystal structure of SARS-CoV-2 S glycoprotein complexed with ACE2 was obtained from the protein data bank (PDB ID: 7A98 [14]). The top mutations in the S protein based on the heatmap were mapped using Biovia Discovery studio visualizer 2020. To assess the effect of noted RBD mutations on ACE2 binding, the crystal structure of the SARS-CoV-2 spike RBD domain complexed with ACE2 (PDB ID: 6LZG [15]) was used. For assessment of the noted mutations on binding to neutralizing antibodies, the SARS-CoV-2 spike RBD domain complexed with two selected mAbs REGN10933/P2B-2F6 were retrieved (PDB ID: 6XDG; resolution 3.90A and 7BWJ; resolution 2.65 Å, respectively) [16,17]. Point mutations were carried out using Biovia Discovery studio visualizer 2020 and the structures of the complexes were subjected to energy minimization using the macro model tool in Schrodinger 2020 using default parameters. The molecular interactions between the RBD–ACE2 interface, within the RBD and between the neutralizing mAbs-RBD were analyzed using a non-bonded interactions tool in Biovia Discovery studio visualizer 2020. The structure of the S protein of Wuhan-Hu-1 (Genbank ID: YP_009724390.1) inclusive of the pre-cleavage motif PRRAR was modeled based on homology modeling using SWISS-MODEL (https://swissmodel.expasy.org/, Accessed on 31 May 2021) with the template as a SARS-CoV-2 spike protein trimer complex, 7CWU.pdb. 

## 3. Results

The genomic surveillance for the spurt in the COVID-19 cases in Maharashtra was carried out to identify the circulating lineages/VOCs and identify possible functionally significant mutations in the spike protein.

PangoLIN lineage classification of 1791 whole genomes revealed the presence of 40 lineages, with the majority within GISAID clades G/GH/GR/GRY/GV, evident from the D614G mutation in the S protein. Lineages B.1.617.1 (*n* = 779), B.1.617.2 (*n* = 478), B.1.1.306 (*n* = 116), B.1.36.29 (*n* = 100), B.1.1.7 (*n* = 75), B.1.617.3 (*n* = 51) were found to be the predominant lineages (Figure 1 and Appendix A). Chronologically, the earliest detected samples of lineage B.1.617.1, B.1.617.2 and B.1.617.3 were observed in January 2021, December 2020 and February 2021, respectively (Figure 2). The three new lineages, B.1.617.1, B.1.617.2 and B.1.617.3, could be linked to mutations specific to the spike region, along with ORF1a, ORF1b, ORF3a, M, ORF7a and N (Figure 3 and Appendix A). 

Among the new B.1.617 lineages, B.1.617.1 included the majority of the strains from eastern part of Maharashtra while B.1.617.2 also included sequences from major cities like Pune, Thane and Mumbai in the western part of the state. The mutations L452R and E484Q within the RBD were specific to lineage B.1.617.1 and B.1.617.3 while L452R and T478K were specific to lineage B.1.617.2. The lineage B.1.617.3 was characterized by mutations T19R and E484Q. Mutation G142D and P681R, within the spike but outside the RBD region, were common to all the three new lineages (Figure 3). P681R is noted in the S1-S2 furin cleavage site. Specific mutations such as T95I, H1101D and D1153Y were found in a proportion of B.1.617.1. Similarly, mutations K77T and A222V were found in a proportion of B.1.617.2. A synonymous mutation D111D was observed to be co-occurring with the RBD mutations L452R and E484Q in lineage B.1.617.1. Deletions 157/158 and deletions 119/120 with respect to lineage B.1.617.2 were noted.

The key mutations in the S protein are mapped on a furin-cleaved structure of the S protein (Figure 4). The structural implications of the RBD mutations, L452R and E484Q, as in lineages B.1.617.1 and B.1.617.3, were analyzed in terms of interaction with the ACE2 receptor and neutralizing antibodies that are known to have interactions with these residues (Appendix A). Similarly, the implications of RBD mutations, L452R and T478K, as in lineage B.1.617.2, were studied. The effect of the E484Q mutation is noted in terms of disruption in an electrostatic bond of the spike RBD residue E484 with K31 in the ACE2 interaction interface (Figure 5AAppendix A). The intramolecular interactions in the wild strain indicate that the L452 residue is involved in a hydrophobic interaction with L492 which forms another hydrophobic contact with F490. These residue interactions form a hydrophobic patch on the surface of the RBD. The L452R mutation abolishes the hydrophobic interaction with L492 of the RBD. Estimation of the minimized energies of the wildtype and L452R occurring with the E484Q mutant structure of the RBD complexed with ACE2 showed energy values of −93732.305 kcal/mol and −94543.180 kcal/mol, respectively. In the case of the mutant RBD possessing L452R with T478K (Figure 5B), though no additional intermolecular contact was affected, three additional hydrogen bonds (H-bonds) were formed by K478 with F486. The intramolecular interaction L452–L492 associated with the L452R mutation was disrupted before. The minimum energy value of the RBD–ACE2 complex was 94,751.367 kcal/mol.

The mutations L452R and E484Q are seen to disrupt the interfacial interactions of the spike RBD with specific neutralizing antibodies (Figure 5C,D; Appendix A), The heavy chain of monoclonal antibody REGN10933 interacts with the RBD by making two H-bonds between E484 of the RBD and Y53 and S56 of the antibody. Residue Y453 of the RBD makes an H-bond with D31 of the heavy chain of the antibody. The RBD mutation E484Q disrupts the two H-bonds with S56 and Y53 (Figure 5C). Residues L452 and E484 possess direct contacts with the mAb P2B-2F6 (Figure 5D). The intermolecular interactions of the wildtype complex indicate that L452 is involved in hydrophobic interactions with residues I103 and V105 of the heavy chain of the antibody. Residue E484 forms H-bonds with N33 and Y34 of the light chain as well as a salt bridge and H-bond interaction with the R112 side chain of the light chain. The L452R mutation breaks the hydrophobic interactions with both residues I103 and V105 and also disrupts the H-bond and electrostatic interaction with R112.

Modeling the S protein with the pre-cleavage motif PRRAR showed the exposure of R682 and R685 (Appendix A). Induction of a P681R point mutation in the modeled structure further revealed that the side chain of R681 could facilitate additional interactions with furin.

## 4. Discussion

In addition to the three global VOCs, the global variants of interest (VOIs) such as GH/452R.V1/ B.1.427/B.1.429 (epsilon), GR/P.2 (zeta), G/484K.V3/B.1.525 (eta), GR/P.3 (theta) and GH/B.1.526 (iota) have been identified by the WHO [6]. The potential consequences of emerging variants are increased transmissibility, increased pathogenicity and the ability to escape natural or vaccine-induced immunity [18,19]. With this background, we sequenced and analyzed the whole genomes of SARS-CoV-2 from different districts of Maharashtra where a surge in COVID-19 activity was noted since the end of January 2021 after a gap of almost four months.

Phylogenetic and sequence analyses revealed that only a small proportion of sequences, mostly in the month of December 2020, were detected as the VOC “alpha”. The deletions 69/70 and 144/145 specific to this lineage were noted in these strains. On the other hand, the B.1.617.1 lineage possessing common signature mutations L452R, E484Q and P681R in the spike protein could be linked to the surge of cases in February 2021 in eastern Maharashtra. In addition, districts in western Maharashtra, such as Pune, Mumbai, Thane and Nashik, showed the presence of multiple lineages in circulation in comparison to the dominance of lineage B.1.617.1 in eastern Maharashtra (Appendix A). From March 2021, lineage B.1.617.1 declined and was overtaken by B.1.617.2. The epidemiological evidence with respect to the surge being noted in the eastern districts suggests that B.1.617.1 has its epicenter in that part of Maharashtra. Further, movements associated with social gatherings and local body elections most likely led to the spread to the western parts of the state. This is where major highly populous metropolitan cities are located.

A comparison of the lineages circulating in India during the study period from November 2020 to May 2021 (Appendix A), revealed that from among the new lineages, only B.1.617.1 and B.1.617.2 were the predominant ones in circulation. Analyzing retrospectively, there were single sequences found to be submitted to GISAID of lineage B.1.617.1 in December 2020 from Maharashtra (district Nagpur) and Odisha. Similarly, B.1.617.2 was noted in September 2020 from the state of Madhya Pradesh, in November 2020 from Uttar Pradesh and in December in Bihar and Maharashtra. The proportion of B.1.617.1 and B.1.617.2 in Maharashtra from February–March 2021 was found to be about 55–60% and 10–60%, respectively, as compared to 5 to 15% and 3–10% in the rest of the country (Appendix A). Less sequencing before the study period may have resulted in missing out on earlier detections of these lineages. Overall, the observations may suggest the dispersal of the new lineages from Maharashtra to the other states in India. The early detection of the variants in the states of Uttar Pradesh, Bihar and Odisha could be attributed to the movement of migrant workers from the state of Maharashtra. In addition, in India during the period from November 2020 to May 2021, there were other lineages, such as B.1, B.1.1, B.1.36, B.1.351, B.1.618 and B.1.1.216, that were not common in the state of Maharashtra. Our earlier studies [9,20] that were based on genome sequences from India from January to August/September 2020 had demonstrated that the dominant lineages were B.1.113, B.1.1.32, B.1.1.8, B.1.80, B.4 and B.6. This reflects the diversity of lineages in the country and also the switches in the dominant lineages.

Notably, D111D was found to be associated with the signature mutations of lineage B.1.1.617.1. The co-occurrence of synonymous mutations with the non-synonymous mutations observed is interesting and generally being reported [21,22].

The structural analysis of the effect of RBD mutations L452R and E484Q towards ACE2 binding revealed a decrease in intramolecular and intermolecular contacts with respect to the wildtype. However, the hydrophobic L452 residue mutation to the hydrophilic 452R might help in interactions with water molecules and overall stabilization of the complex, as was reflected in the lower minimum energy of the mutant complex. The effect of the mutations L452R and T478K on ACE2 binding was also observed as enhanced stabilization of the RBD–ACE2 complex.

Another significant mutation, P681R, in the furin cleavage site resulted in enhancement of the basicity of the poly-basic stretch, and the likely facilitation of additional contacts with furin for S1–S2 cleavage. This could help in an increased rate of membrane fusion, internalization and thus better transmissibility. A recent study [23] revealed increased infectivity of the B.1.617 spike protein that could be attributed to L452R which itself caused a 3.5-fold increase in infectivity and, in combination with E484Q, caused a 3-fold increase. The P681R mutation caused a small increase in proteolytic processing that might have an effect on infectivity [24]. Observations on enhanced transmissibility of B.1.617.2 are also reported [23]. Further in vitro and/or in vivo studies to understand the phenotypic effects of the mutant strains would be vital for validating these observations.

Structural analysis further showed that the two RBD mutations L452R and E484Q may decrease the binding ability of REGN10933 and P2B-2F6 antibodies to the variant strains, compared to that in the wildtype strain. A recent report [25] revealed that the L452R mutation reduced or abolished neutralizing activity of 14 out of 35 RBD-specific mAbs including three clinical stage mAbs. Another recent study [26] has demonstrated that the mutation L452R can escape from human leukocyte antigen (HLA) 24-restricted cellular immunity and can also increase viral infectivity, potentially promoting viral replication. The combination of the two RBD mutations, L452R and E484Q, noted in this study could affect the neutralization of the select mAbs. The neutralizing potential of a wider range of mAbs against the strains of the B.1.617 lineages needs to be assessed. Recent studies have investigated the level of neutralization efficacy of sera from vaccinees of different vaccine formulations and those who have experienced natural infection. The neutralization of B.1.617.1 with sera of BBV152 (Covaxin) vaccinees showed almost two-fold reduction in neutralization compared to a B.1 “G” clade strain [27]. Even more reduction in neutralizing titer with sera of ChAdOx1 2 (Covishield)-vaccinated individuals was observed against the B.1.617.1 variant [28]. Against the B.1.617.2 variant, a 4.6-fold and 2.7-fold reduction in the neutralization titer was observed in comparison to the B.1 variant with sera of COVID-19 recovered cases and Covaxin vaccinees, respectively [29]. Bernal et al. [30], have also reported the effectiveness of COVID-19 vaccines against the B.1.617.2 variant. A summary of the major spike protein mutations defining the B.1.617 lineages and their proposed effects [31,32,33,34,35,36,37] is provided in Appendix A.

Mutations at both the residue positions 452 and 484 individually have been reported previously. L452R has been noted earlier in lineages B.1.427 and B.1.429 while the E484K mutation is common to the two VOCs beta and gamma and two VUIs, zeta and eta. E484Q has also been reported in several sequences in the GISAID. P681H is one of the mutations in the VIC alpha, while P681R is one of the mutations in the lineage A.23.1, which is identified as a “variant under monitoring” (https://www.ecdc.europa.eu/en/COVID-19/variants-concern, Accessed on 14 June 2021). The new lineages B.1.617.1 and B.1.617.3 in this study are thus a unique combination of spike mutations L452R, E484Q and P681R. Between these two lineages, it was observed that B.1.617.1 spread more rampantly. The T478K mutation in the spike protein in lineage B.1.617.2 has been seen in Mexican variant B.1.1.222. This mutation or its combination with the other mutations in the genome appears to give a greater fitness advantage to the B.1.617.2 lineage as this lineage took over from B.1.671.1, and became dominant in Maharashtra and India. As of June 12, 2021, the worldwide prevalences of B.1.617.1 and B.1.617.3 are <0.5% while the prevalence of B.1.617.2 is 2% (outbreak.info). As per GISAID submissions, the B.1.617.2 lineage has been reported from 64 countries including the UK, USA, Germany, Singapore, Japan, Australia, etc. The global spread of the lineage resulted in its designation as a VOC delta. Lineage B.1.617.1 has been assigned as a VOI kappa [6].

To summarize, the study investigated the S protein mutations associated with the rise in COVID-19 cases in Maharashtra observed since the month of February 2021. The continuous increase in positivity could be attributed to signature mutations in the spike protein and functionally significant co-occurring mutations. The investigations related to patterns of circulation of the new variant lineages during the COVID-19 second wave in India need to be continued for their public health impact and association with vaccine breakthroughs, both in the case of partially and fully vaccinated individuals and reinfections.

## Figures and Tables

**Figure 1 microorganisms-09-01542-f001:**
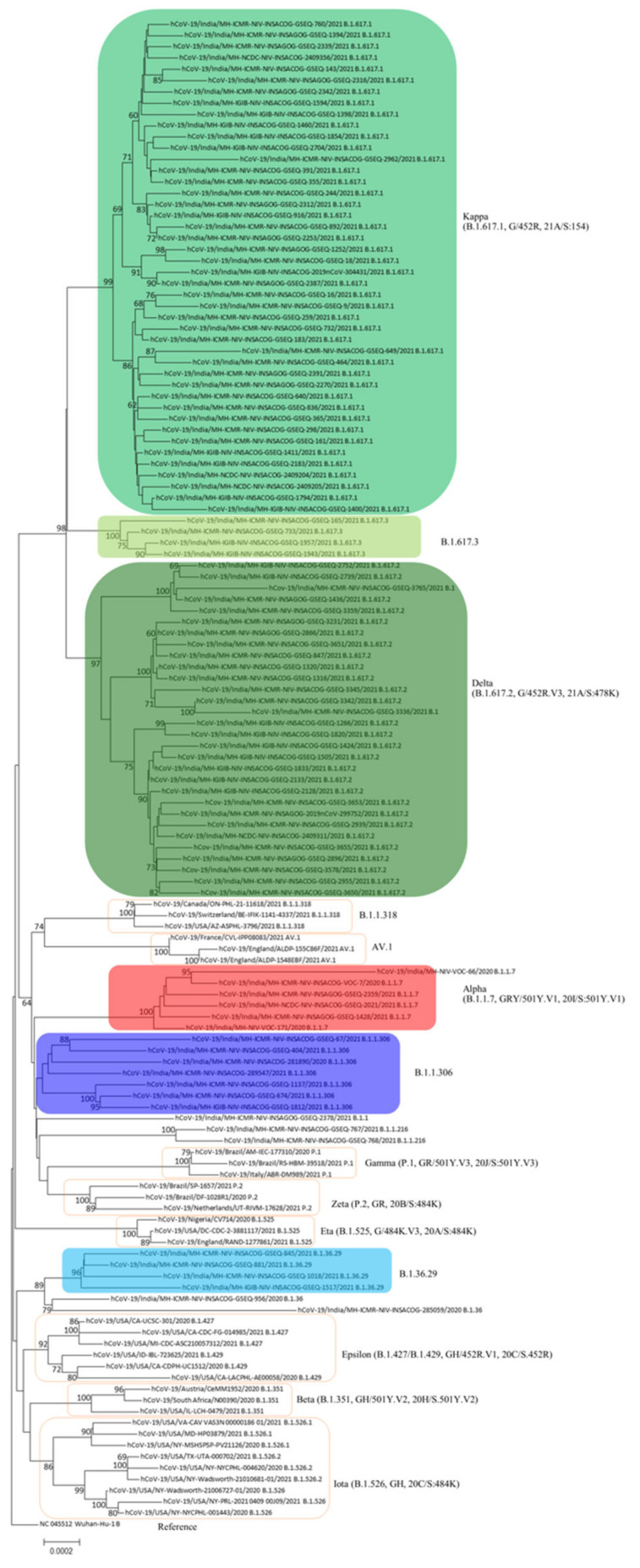
Neighbor-joining tree of representative SARS-CoV-2 genomes depicting lineages in WHO label, PangoLIN, GISAID and Nextstrain. The sequences of this study are tagged as India/MH-ICMR-NIV and major lineages are in colored boxes. Additionally, included are representatives of the global WHO identified VOCs/VOIs/VUIs in unfilled boxes.

**Figure 2 microorganisms-09-01542-f002:**
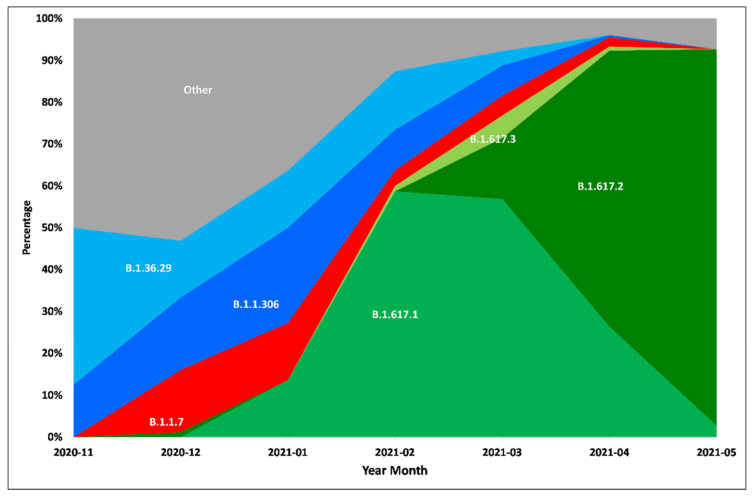
Temporal trend of major lineages in the districts of Maharashtra from November, 2020 to May, 2021.

**Figure 3 microorganisms-09-01542-f003:**
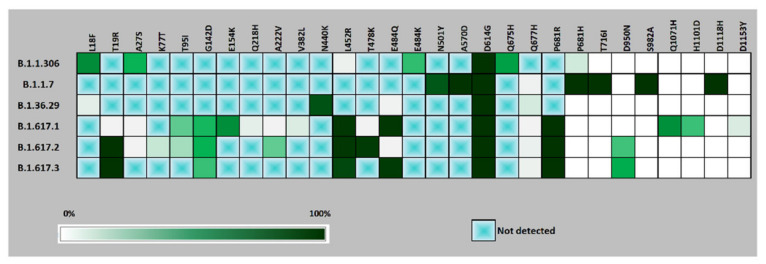
Heat map of major mutations in the spike protein of the predominant lineages from November, 2020 to May, 2021. A box with a green cross indicates that the specific mutation was absent.

**Figure 4 microorganisms-09-01542-f004:**
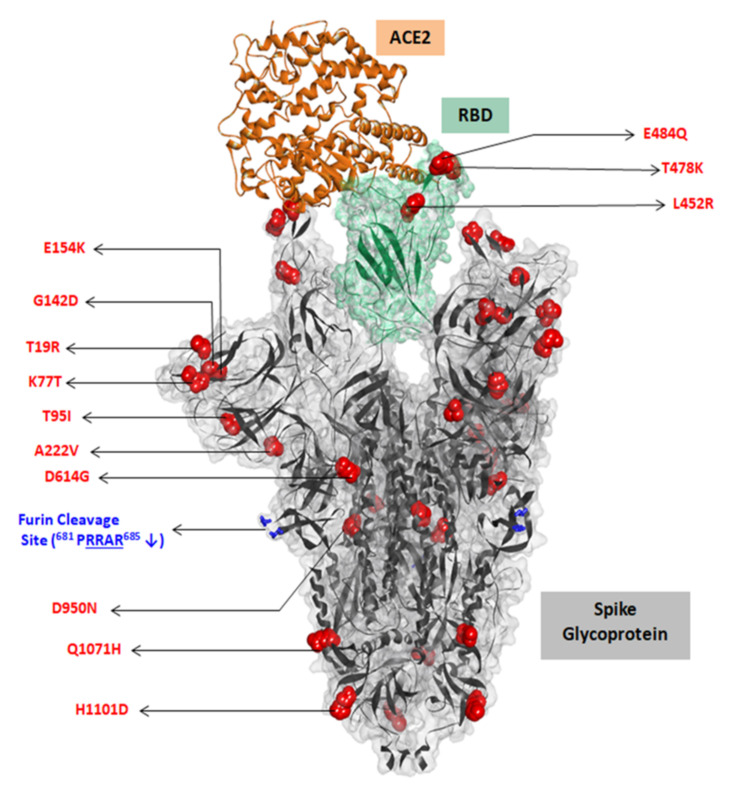
Mapping of key mutations on the furin-cleaved crystal structure of SARS-CoV-2 spike glycoprotein (gray surface view) in complex with ACE2 (solid brown ribbon). RBD region shown in green.

**Figure 5 microorganisms-09-01542-f005:**
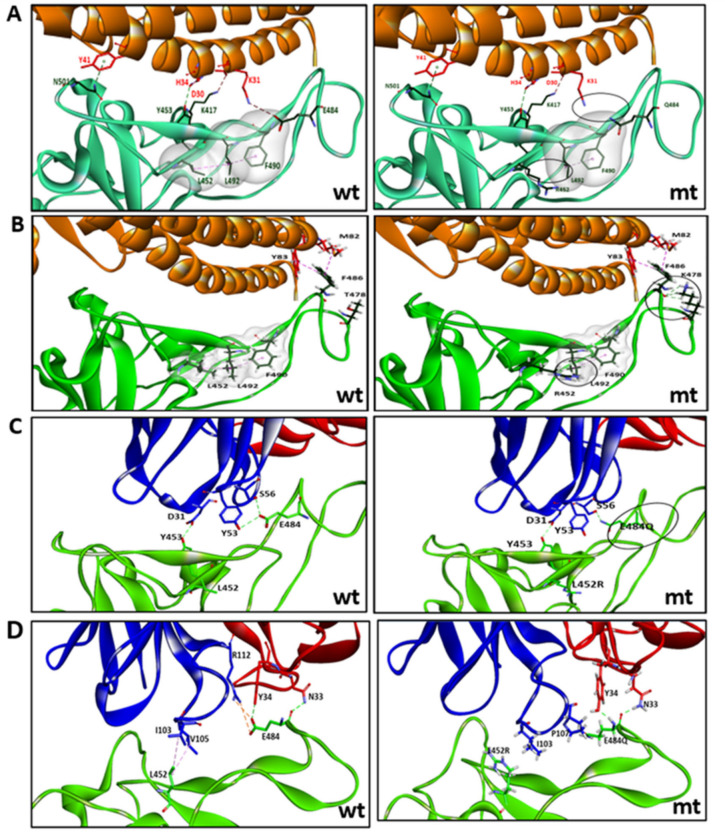
(**A**) Key interactions between ACE2–RBD involving mutations L452R and E484Q in the RBD, as in lineages B.1.617.1 and B.1.617.3. (**B**) Key interactions between ACE2–RBD involving mutations L452R and T478K in the RBD, as in lineage B.1.617.2. (**C**) Interactions between RBD–mAb REGN10933. (**D**) Interactions between RBD–mAb P2B-2F6. wt, corresponds to wildtype strain and mt, mutant strain. In (**A**) and (**B**) are the intra-molecular contacts in a hydrophobic patch of the RBD region (surface displayed in gray). In (**C**) and (**D**), blue represents heavy chain and red represents light chain.

## Data Availability

Genomes sequenced as a part of this study have been deposited in GISAID, with INSACOG tags.

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
