# Peer review of "SARS-CoV-2 Spike Mutations, L452R, T478K, E484Q and P681R, in the Second Wave of COVID-19 in Maharashtra, India"

_microorganisms, 2021, doi:10.3390/microorganisms9071542_

Round 1

Reviewer 1 Report

In this manuscript, the authors sequenced and analysed 598 SARS-CoV-2 complete genomes collected in Maharashtra, India. They described the distribution of the most frequent substitutions in the spike protein across major circulating lineages and in time, and also mapped these substitutions onto the PDB structure of S protein. Overall, this is a solid addition to the available SARS-CoV-2 data coming out of this epidemically important country, which merits publication. The comments below are structured as major, minor and suggestions; suggestions will (in my view) increase the value of the manuscript but are not essential for publication.

MAJOR ISSUES

  1. Please clarify how “the top ten mutations” were selected - based on the highest total frequency or based on the most pronounced frequency growth in the past three months?
  2. It is unclear what is depicted by the dark blue line in Fig. 2. Is this the number of nonsynonymous difference from reference for the sequences collected in this month, the number of novel nonsynonymous mutations that were not observed previously, the overall number of observed nonsynonymous mutations, or something else? Please clarify.
  3. Please add some measure of statistical support, e.g. bootstrap support values, to branches in Figure 1, at least for the major branches that you discuss (B.1.617.A, B.1.617.B, etc.)
  4. Line 154 - whichmutationsexactlywereintroducedintothemutantstructure?

MINOR ISSUES

  1. Figures 2 and 3 are mislabeled in the text.
  2. Figure 5A - both panels are labeled ‘wt’

SUGGESTIONS

  1. The authors limited their analysis to genomes sequenced in the study. It would help to see how these mutations and lineages are distributed in all publicly available Indian sequences.
  2. It would be interesting to see the data on the most abundant deletions.
  3. The phylogeny in Fig. 1 would be much more useful if the actual branch lengths were shown.
  4. It would help to make the structural part of the manuscript more structured (e.g. in the form of a table where for each position discussed would be it’s proposed effect on the structure and references if available)
  5. How abundant are combinations of the discussed substitutions in lineages circulating outside India?

Author Response

Reply to comments and suggestions for authors

Reviewer-1

In this manuscript, the authors sequenced and analysed 598 SARS-CoV-2 complete genomes collected in Maharashtra, India. They described the distribution of the most frequent substitutions in the spike protein across major circulating lineages and in time, and also mapped these substitutions onto the PDB structure of S protein. Overall, this is a solid addition to the available SARS-CoV-2 data coming out of this epidemically important country, which merits publication. The comments below are structured as major, minor and suggestions; suggestions will (in my view) increase the value of the manuscript but are not essential for publication.

 MAJOR ISSUES

  1. Please clarify how “the top ten mutations” were selected - based on the highest total frequency or based on the most pronounced frequency growth in the past three months?

Response: It was based on the highest total frequency. The same has been clarified now in the methods section. In addition, based on the suggestion of Reviewer 3, we have now mapped the genomic mutations of the top six lineages using a heat map (Figure 3 & Fig. S2).

  1. It is unclear what is depicted by the dark blue line in Fig. 2. Is this the number of nonsynonymous difference from reference for the sequences collected in this month, the number of novel nonsynonymous mutations that were not observed previously, the overall number of observed nonsynonymous mutations, or something else? Please clarify.

Response: At the time of revision of this manuscript, on the basis of marker mutations three new lineages have now been identified in the Pango lineage nomenclature. Hence, this figure has been replaced by a new figure2 depicting the temporal trends of the new lineages.

  1. Please add some measure of statistical support, e.g. bootstrap support values, to branches in Figure 1, at least for the major branches that you discuss (B.1.617.A, B.1.617.B, etc.)

Response: We have now reconstructed the phylogeny based on genetic distances as was also suggested by reviewers 2 & 3. The new figure 1, shows the bootstrap values. The additions to methodology section have been made accordingly.

  1. Line 154 – which mutations exactly were introduced into the mutant structure?

Response: The mutations have now been clarified in the Materials & Methods, results and discussion sections. In addition, we have now modelled the T478K mutation along with the L452R mutation as is present in lineage B.1.617.2. The results of the effect of these mutations on ACE2 binding are shown in Fig. 5B.  The necessary additions in results and discussion section of the revised manuscript have been made.

MINOR ISSUES

  1. Figures 2 and 3 are mislabeled in the text.

Response: Sorry about the mix up in the labels. However based on the updated analyses, new figures have now been incorporated.

  1. Figure 5A - both panels are labeled ‘wt’
  1. Response: Sorry for the over sight. The necessary change has been incorporated in fig. 5.

SUGGESTIONS

  1. The authors limited their analysis to genomes sequenced in the study. It would help to see how these mutations and lineages are distributed in all publicly available Indian sequences.

Response: We have now also carried out similar analysis of the whole genomes from India as available in GISAID. The results of this have been discussed. New supplementary figure S5 has been incorporated.

  1. It would be interesting to see the data on the most abundant deletions.

Response: The most abundant deletions in the lineages are as per the lineage markers. The same are mentioned in the revised manuscript discussion section.

  1. The phylogeny in Fig. 1 would be much more useful if the actual branch lengths were shown.

Response: Based on this suggestion, we have now constructed a neighbor joining tree (Figure 1).

  1. It would help to make the structural part of the manuscript more structured (e.g. in the form of a table where for each position discussed would be it’s proposed effect on the structure and references if available)

Response: As per the suggestion we have now included a Supplementary table 3 with the details regarding the effect of the mutations, the lineage and the references.

  1. How abundant are combinations of the discussed substitutions in lineages circulating outside India?

Response: Based on the global dataset of GISAID sequences and the information available in Outbreak.info, the cumulative prevalence of lineages B.1.617.1, B.1.617.2 and B.1.617.3, are now incorporated in the discussion section.

Reviewer 2 Report

More than one year after the start of the SARS-CoV-2/COVID-19 pandemic, several SARS-CoV-2 variants have emerged in multiple locations worldwide. With respect to mutations in the viral genomes, most attention is given to the spike glycoprotein that is required for cell entry and is the main target for neutralizing antibodies. While each SARS-CoV-2 variant is defined by so-called signature mutations, some variants share some mutations, leading to the question, whether these mutations are due to an identical origin (same progenitor virus, lineage) or whether they evolved independently (convergent evolution, homoplasy).

In this study, Cherian and coworkers analyzed SARS-CoV-2 isolates from Maharashtra/India, including variant B.1.617. In particular, the authors focused on mutations L452R, E484Q and P681R that have been found also in other SARS-CoV-2 variants: mutation L452R is also found in variants B.1.427 and B.1.429; mutation E484Q is also found in viruses from different lineages; P681R is also found in A.23.1. Based on their phylogenetic analyses the authors conclude that the combined occurrence of L452R/E484Q/P681R was a result of convergent evolution. The authors further analyzed the impact of the receptor binding domain (RBD) mutations on spike-ACE2 interaction and binding of monoclonal antibodies in silico. Here the authors found that L454R and E484Q will likely decrease binding of antibodies REGN10933 and P2B-2F6 while both mutations reduce intra- and intermolecular contacts regarding ACE2 binding. With respect to mutation P681R at the S1/Se cleavage site the authors speculate that the increased basicity of the cleavage site due to P681R may enhance membrane fusion.

Although, independent acquisition of identical mutations in distinct SARS-CoV-2 lineages may reflect convergent evolution (e.g. due to the same selective pressures and limited evolutionary capacity to cope with them), the overall limited genetic diversity in SARS-CoV-2 makes this kind of analysis very challenging. Especially, given the potential bias by phylogenetic analyses (see major points). In order to support the phylogenetic data (and thus the main conclusion) it is crucial to validate the data.

While the manuscript itself is well written, some data have been already published by others and the main conclusion is not fully supported by the data. The main weaknesses of the manuscript are (i) the lack of novelty (reduced binding of REG10933 due to L452R and E484Q has already been reported by others, including in vitro confirmation) (ii) missing data to support their conclusions (no scientific evidence supporting the speculation that P681R enhances membrane fusion) and (iii) no validation of the phylogenetic data.

Major points

  • In their recent preprint, Jo and coworkers argue that convergent evolution in SARS-CoV-2 cannot be reliably inferred from phylogenetic analyses (doi: https://doi.org/10.1101/2021.05.15.444301). Mainly due to the limited genetic diversity in SARS-CoV-2 and simple nature of the mathematical models that are the basis for phylogenetic analyses. Usually the tree with the highest likelihood is selected but other trees with only slightly lower likelihood but different phylogenetic relationship between virus isolates are not considered. Have the authors tested whether alternative phylogenetic trees with lower (but still high likelihood) come to the same result? Are the authors sure that this phylogenetic tree is the “correct one”. If yes, please specify how the data were validated.
  • The phylogenetic analysis (and its validation, if done) needs to be described in more detail.

Minor points

  • Line 35: Meanwhile, the WHO has declared B.1.617 a VOC
  • Line 44: ACE2 not ACE21 (maybe misplaced reference?)
  • Please do not refer to variants B.1.1.7, B.1.351 and P.1 as the “UK“, “South Africa” and “Brazil” variants. Fist, stigmatization should be avoided. Second, although for example the B.1.1.7 variants has been first identified in the UK it is now dominant in many countries and not only the UK.
  • Line 102: “Wuhan” is not the correct description of the reference strain (Wuhan-Hu-1). Please also add an identifier (GenBank or GISAID ID) in the methods section.
  • Line 132: “Figure 2” should be “Figure 3”
  • Line 142: “Figure 3” should be “Figure 2”

Author Response

Reviewer 2

More than one year after the start of the SARS-CoV-2/COVID-19 pandemic, several SARS-CoV-2 variants have emerged in multiple locations worldwide. With respect to mutations in the viral genomes, most attention is given to the spike glycoprotein that is required for cell entry and is the main target for neutralizing antibodies. While each SARS-CoV-2 variant is defined by so-called signature mutations, some variants share some mutations, leading to the question, whether these mutations are due to an identical origin (same progenitor virus, lineage) or whether they evolved independently (convergent evolution, homoplasy).

In this study, Cherian and coworkers analyzed SARS-CoV-2 isolates from Maharashtra/India, including variant B.1.617. In particular, the authors focused on mutations L452R, E484Q and P681R that have been found also in other SARS-CoV-2 variants: mutation L452R is also found in variants B.1.427 and B.1.429; mutation E484Q is also found in viruses from different lineages; P681R is also found in A.23.1. Based on their phylogenetic analyses the authors conclude that the combined occurrence of L452R/E484Q/P681R was a result of convergent evolution. The authors further analyzed the impact of the receptor binding domain (RBD) mutations on spike-ACE2 interaction and binding of monoclonal antibodies in silico. Here the authors found that L454R and E484Q will likely decrease binding of antibodies REGN10933 and P2B-2F6 while both mutations reduce intra- and intermolecular contacts regarding ACE2 binding. With respect to mutation P681R at the S1/S2 cleavage site the authors speculate that the increased basicity of the cleavage site due to P681R may enhance membrane fusion.

Although, independent acquisition of identical mutations in distinct SARS-CoV-2 lineages may reflect convergent evolution (e.g. due to the same selective pressures and limited evolutionary capacity to cope with them), the overall limited genetic diversity in SARS-CoV-2 makes this kind of analysis very challenging. Especially, given the potential bias by phylogenetic analyses (see major points). In order to support the phylogenetic data (and thus the main conclusion) it is crucial to validate the data.

While the manuscript itself is well written, some data have been already published by others and the main conclusion is not fully supported by the data. The main weaknesses of the manuscript are (i) the lack of novelty (reduced binding of REG10933 due to L452R and E484Q has already been reported by others, including in vitro confirmation) (ii) missing data to support their conclusions (no scientific evidence supporting the speculation that P681R enhances membrane fusion) and (iii) no validation of the phylogenetic data.

Major points

In their recent preprint, Jo and coworkers argue that convergent evolution in SARS-CoV-2 cannot be reliably inferred from phylogenetic analyses (doi: https://doi.org/10.1101/2021.05.15.444301). Mainly due to the limited genetic diversity in SARS-CoV-2 and simple nature of the mathematical models that are the basis for phylogenetic analyses. Usually the tree with the highest likelihood is selected but other trees with only slightly lower likelihood but different phylogenetic relationship between virus isolates are not considered. Have the authors tested whether alternative phylogenetic trees with lower (but still high likelihood) come to the same result? Are the authors sure that this phylogenetic tree is the “correct one”. If yes, please specify how the data were validated.

Response: Based on the comment of this and other referees, we have now repeated the phylogenetic analyses. We agree with the recent work by Jo & coworkers as referred by you. Based on this and the comment of the referee 3, we have refrained from claiming that there is convergent evolution. Thanking you for your valuable input.

The phylogenetic analysis (and its validation, if done) needs to be described in more detail.

Response: The phylogenetic analysis has been repeated based on the comments of this and the other reviewers. An up-to-date data set of the sequences generated for the state of Maharashtra was used. The same has now been described in greater detail in the materials & methods section.

Minor points

Line 35: Meanwhile, the WHO has declared B.1.617 a VOC

Response: Yes, we have now made the necessary changes in the abstract and the discussion section.

Line 44: ACE2 not ACE21 (maybe misplaced reference?)

Response: Yes. It has now been corrected.

Please do not refer to variants B.1.1.7, B.1.351 and P.1 as the “UK“, “South Africa” and “Brazil” variants. Fist, stigmatization should be avoided. Second, although for example the B.1.1.7 variants has been first identified in the UK it is now dominant in many countries and not only the UK.

Response: Agree and thank the referee for this. We have avoided the reference to the variants in this form throughout the text.

Line 102: “Wuhan” is not the correct description of the reference strain (Wuhan-Hu-1). Please also add an identifier (GenBank or GISAID ID) in the methods section.

Response: Done.

Line 132: “Figure 2” should be “Figure 3” & Line 142: “Figure 3” should be “Figure 2”

Response: Sorry about the mix up in the labels. However based on the updated analyses, new figures have now been incorporated.

Reviewer 3 Report

The study by Cherian et al investigates the emergence of B.1.617 in India. This is a new SARS-CoV-2 variant who might have increased transmissibility. The authors have sequenced >500 full-length SARS-CoV-2 genomes and report on their genetic diversity and phylogeny, as well as how the different mutations might affect the protein structure and subsequent binding affinity. I don't have enough knowledge in protein structure and binding to assess that part of the paper, although, the methods and results seem adequate to me. I believe the paper is mostly well written and well presented, however, the phylogenetic presentation could be improved. At the moment, it is not clear to me if convergence really occurred or if the variants containing the mutations have a common origin. Minor comments: Phylogeny: it might be worthwhile adding some global data, sequences from other countries containing the named variants to emphasise the convergence of these mutations. I would expect such sequences to fall basal to the tree and form independent nodes to B.1.617. How did the authors define clusters/clades? Bootstrap value? Figure 1. The tree layout is a bit confusing, it looks like the branch length was normalised. I think a rectangular layout branches representing nucleotide substitutions per site would be more informative. It would be easier to see the genetic distances between lineages as well as which nodes contain the same mutations. Maybe the authors could map the genome with the mutations next to each clade. Would it be possible to annotate geographic regions to the three B.1.617 clades? It is interesting how these have separated, what could be the reason? The authors overuse the word 'noted' which should be replaced with more suitable and specific terms for the context. For example line 142 'observed' would be a better choice. Line 149. 'wild' do the authors mean 'wild-type'? if so what strain? Something seems wrong with the reference annotations in the manuscript. Should the numbers be superscript?

Author Response

Reviewer 3

The study by Cherian et al. investigates the emergence of B.1.617 in India. This is a new SARS-CoV-2 variant that might have increased transmissibility. The authors have sequenced >500 full-length SARS-CoV-2 genomes and report on their genetic diversity and phylogeny, as well as how the different mutations might affect the protein structure and subsequent binding affinity. I don't have enough knowledge in protein structure and binding to assess that part of the paper, although, the methods and results seem adequate to me.

I believe the paper is mostly well written and well presented, however, the phylogenetic presentation could be improved. At the moment, it is not clear to me if convergence really occurred or if the variants containing the mutations have a common origin.

Minor comments:

Phylogeny: it might be worthwhile adding some global data, sequences from other countries containing the named variants to emphasise the convergence of these mutations. I would expect such sequences to fall basal to the tree and form independent nodes to B.1.617.

Response: The phylogenetic analysis has been repeated based on the comments of this and the other reviewers and with an updated dataset. The representatives of the global VOCs were also included. As no clear evidence of convergence was observed and also based on the reference (Jo et al.) suggested by the earlier reviewer, we have now modified the text accordingly.

How did the authors define clusters/clades? Bootstrap value?

Response: The assignment of clades is as per the new nomenclature available in the PangoLIN classification system and the clades also have good bootstrap values.

Figure 1. The tree layout is a bit confusing, it looks like the branch length was normalised. I think a rectangular layout branches representing nucleotide substitutions per site would be more informative. It would be easier to see the genetic distances between lineages as well as which nodes contain the same mutations. Maybe the authors could map the genome with the mutations next to each clade.

Response: Yes considering the suggestion, we have now reconstructed the phylogeny (Figure 1) based on genetic distances as was also suggested by reviewer 2. We have also now mapped the genomic mutations of the top six lineages using a heat map (Figure 3 & Figure S2).

Would it be possible to annotate geographic regions to the three B.1.617 clades? It is interesting how these have separated, what could be the reason?

Response: Figure S1 now shows the district-wise distribution of the three B.1.617 clades. The possible reason for the geographical separation of the two lineages is also mentioned in discussion section.

The authors overuse the word 'noted' which should be replaced with more suitable and specific terms for the context. For example line 142 'observed' would be a better choice.

Response: Thank you. We have tried to replace with more suitable and specific terms.

Line 149. 'wild' do the authors mean 'wild-type'? if so what strain?

Response: Yes wild-type strain. The Wuhan-Hu-1 strain was used as wild-type.

Something seems wrong with the reference annotations in the manuscript. Should the numbers be superscript?

Response: Yes, we have made the reference annotations as superscripts.

Reviewer 4 Report

In general I like this communication.

1) My only concern is related to statistical representativeness of those 733 genomes that were proceeded.

2) I would like to see some words about this in Materials and Methods on p. 2. I would also like to see comparison of results that were obtained in Sarkar et al. (2021) and based on genomes of  837 SARS-CoV-2 viruses circulating in India in March-August 2020 with the presented results that are based on 733 SARS-CoV-2 viruses circulating in India in November 2020 - March 2021.

3) Citations should be formatted as superscripted. Reference 18 Garcia-Beltran is not valid. Ref. 20 is still under the review process.

Author Response

Reviewer 4

In general I like this communication.

1) My only concern is related to statistical representativeness of those 733 genomes that were processed. I would like to see some words about this in Materials and Methods on p. 2.

Response: The dataset has been further updated to include additionally sequenced genomes over the period upto May 2021. Both eastern (n=829) and western (n=962) districts of the state are well represented. The updated dataset has also ensured the representativeness of the samples based on the size of the outbreaks in the districts. This is also mentioned in the materials & methods section.

2) I would also like to see comparison of results that were obtained in Sarkar et al. (2021) based on genomes of  837 SARS-CoV-2 viruses circulating in India in March-August 2020 with the presented results that are based on 733 SARS-CoV-2 viruses circulating in India in November 2020 - March 2021.

Response: The paper by Sarkar et al., based on 837 genomes of SARS-CoV-2, has classified the genomes based on the old NextStrain nomenclature. In order to compare the results of this study we have now cited our own papers (Potdar et al., 2021 & Yadav et al, 2021). The results of the same which are based on larger data sets over the period upto August/September 2020, are stated in discussion section . In addition, based on the comments of another reviewer, we have also compared with the Indian dataset for the same period as this study ie. Nov. 2020 to May 2021.

3) Citations should be formatted as superscripted.

Response: Done

Reference 18 Garcia-Beltran is not valid.

Response: Complete details of reference now added.

Ref. 20 is still under the review process.

Response: Ref. 20 is published and available in public domain.

Round 2

Reviewer 1 Report

I am happy with the revised version.

Georgii Bazykin (I sign all reviews)

Reviewer 2 Report

The authors adressed all my points.

Minor point:

The resolution of Figure 1 needs to be improved.